# Influence of Single Nucleotide Polymorphism of *ENPP1* and *ADIPOQ* on Insulin Resistance and Obesity: A Case-Control Study in a Javanese Population

**DOI:** 10.3390/life11060552

**Published:** 2021-06-11

**Authors:** Rini Arianti, Nia Lukita Ariani, Al Azhar Muhammad, Ahmad Hamim Sadewa, Arta Farmawati, Pramudji Hastuti, Endre Kristóf

**Affiliations:** 1Laboratory of Cell Biochemistry, Department of Biochemistry and Molecular Biology, Faculty of Medicine, University of Debrecen, H-4032 Debrecen, Hungary; ariantirini@med.unideb.hu; 2Doctoral School of Molecular Cell and Immune Biology, University of Debrecen, H-4032 Debrecen, Hungary; 3Department of Applied Health Sciences, Health Polytechnic Ministry of Health Malang, Malang 65112, Indonesia; nia.ariani@poltekkes-malang.ac.id; 4Department of Nursing Sciences, Health Polytechnic Ministry of Health Ternate, South Ternate 97713, Indonesia; alazharmuhammad08@gmail.com; 5Department of Biochemistry, Faculty of Medicine, Public Health and Nursing, Gadjah Mada University, Yogyakarta 55281, Indonesia; hamimsadewa@gmail.com (A.H.S.); a.farmawati@ugm.ac.id (A.F.); nartyr@ugm.ac.id (S.); pramudji.has@ugm.ac.id (P.H.)

**Keywords:** *ENPP1* gene, *ADIPOQ* gene, circulating adiponectin, obesity, insulin resistance, metabolic syndrome, SNP K121Q (rs1044498), SNP + 276G > T (rs1501299)

## Abstract

Single nucleotide polymorphisms (SNPs) in obesity-related genes, such as ectonucleotide pyrophosphatase phosphodiesterase 1 (ENPP1) and adiponectin (ADIPOQ), potentially increase the risk of insulin resistance, the most common metabolic dysregulation related to obesity. We investigated the association of *ENPP1* SNP K121Q (rs1044498) with insulin resistance and *ADIPOQ* SNP + 267G > T (rs1501299) with circulating adiponectin levels in a case–control study involving 55 obese and 55 lean Javanese people residing in Yogyakarta, Indonesia. Allele frequency was determined by a chi squared test or Fisher’s exact test with an expected value less than 0.05. Odds ratios and 95% confidence intervals were estimated by regression logistic analysis. The presence of the Q121 allele of *ENPP1* resulted in significantly higher fasting glucose, fasting insulin levels, and HOMA-IR, as compared to homozygous K121 carriers. The risk of insulin resistance was elevated in obese individuals carrying Q121 instead of homozygous K121. Adiponectin level was significantly lower in the obese group as compared to the lean group. Obese individuals carrying homozygous protective alleles (TT) of *ADIPOQ* tended to have lower adiponectin levels as compared to GT and GG carriers, however, we did not find statistically significant effects of the +276G > T SNP of the *ADIPOQ* gene on the plasma adiponectin levels or on the development of obesity.

## 1. Introduction

In the last two decades, obesity has shown an increasing prevalence worldwide. Obesity was considered a major health problem only in high income countries, but now it has become a significant health issue in low- and middle-income countries as well. Global data reported that there are 600 million people suffering from obesity [1]. The prevalence of obesity in Indonesian women is higher than in men, both in rural and urban areas [2]. Several factors have contributed to the increased prevalence of obesity in Indonesia, including increase in per capita income, sedentary lifestyle, and dietary habits [3,4].

Obesity significantly increases the risk of type 2 diabetes, atherosclerosis, heart attack, and cancers. Insulin resistance is the most common metabolic dysregulation associated with obesity [5,6,7]. Alterations in the ectonucleotide pyrophosphatase phosphodiesterase 1 (ENPP1) gene were reported as one of the genetic factors involved in obesity and insulin resistance. The human *ENPP1* gene, consisting of 25 exons and 24 introns, is located on chromosome 6q22–23. The encoded protein is a type 2 transmembrane glycoprotein, which is expressed in the adipose tissue and tissues related to glucose and lipid metabolism. Physiological functions of this protein remain unclear, however, ENPP1 is known to be a direct inhibitor of the insulin receptor through interacting with its α-subunit, thereby preventing the conformational change of the insulin receptor [8]. There are several single nucleotide polymorphisms (SNPs) of the *ENPP1* gene. Out of those, the K121Q polymorphism (rs1044498) is the most commonly investigated. The K121Q polymorphism results in a missense mutation in exon 4. Substitution of adenine (A) by cytosine (C) causes the change in the amino acid sequence, lysine (K) to glutamine (Q) at codon 121. The Q variant is a more potent inhibitor of the insulin receptor, since it results in a three-fold stronger interaction with the receptor, compared to the K variant [9]. There were discrepancies regarding the association of the K121Q polymorphism of the *ENPP1* gene with obesity and insulin resistance in various populations. The K121Q polymorphism of the *ENPP1* gene was associated with insulin resistance in a North Indian population [10], however, it was not associated with insulin resistance in Danish Caucasian [11], Japanese [12], and Chinese Han populations [13].

Obesity is also associated with increased adipose cell mass, which leads to elevated adipokine secretion. Adiponectin is one of the adipokines which is secreted at a low level in obese individuals [14,15]. On the other hand, the majority of the adipokines are secreted at a higher level [16]. However, the molecular mechanism behind the decreased adiponectin production in obese individuals remains elusive. Previous studies reported that the alteration of adiponectin level in obese individuals may be triggered by the elevated production of pro-inflammatory cytokines, that can disrupt adiponectin expression at transcriptional or post-translational levels, or adiponectin release into the circulation [17,18]. On the other hand, genetic factors also play a role in determining the variation of plasma adiponectin levels. The +276G > T SNP in the *ADIPOQ* gene (rs1501299) is the most studied in relation to the plasma adiponectin levels. SNP + 276G > T is a G to T nucleotide substitution in intron 2 of the *ADIPOQ* gene. The minor allele of SNP + 276G > T was linked to the alteration of adiponectin levels in previous studies [19,20,21]. Therefore, this SNP might underlie, at least partially, the molecular mechanism which can explain the alterations in adiponectin levels in obese individuals.

The population of Indonesia has a high genetic diversity; thus, investigation of the association of SNPs in *ENPP1* and *ADIPOQ* genes with obesity, and its related traits, is of potential interest. This study aimed to explore the association of the *ENPP1* K121Q (rs1044498) SNP with insulin resistance, and the *ADIPOQ* + 267G > T (rs1501299) SNP with circulating adiponectin levels in a case–control study involving 110 (55 obese and 55 lean individuals) Javanese people residing in Yogyakarta, Indonesia.

## 2. Materials and Methods

### 2.1. Subjects and Ethics Statement

The research design was a case–control study involving 120 subjects, but only 100 subjects were included for *ENPP1* gene SNP examination, and 110 subjects for *ADIPOQ* gene SNP examination. All Javanese subjects were recruited in Yogyakarta. Obese individuals (case group) were defined with a body mass index (BMI) of ≥25 kg/m^2^, and normal subjects were individuals with a BMI of 18–25 kg/m^2^ (control group). Subjects with fasting blood glucose of ≥126 mg/dL, pregnant women, and subjects that received certain drugs (thiazolidinedione, glucocorticoid, angiotensin-converting enzyme, and other antidiabetic drugs) were excluded. All subjects consented to participate in this research approved by the Medical and Health Ethics Committee (MHREC) of the Faculty of Medicine, Gadjah Mada University (reference numbers: KE/FK/278/EC and KE/FK/661/EC).

### 2.2. Blood Collection and Laboratory Measurement

Blood samples from subjects were collected in EDTA vacutainer tubes after a minimum fasting period of 8 h. To separate plasma and buffy coat, total blood was centrifuged at 3500× *g* for 15 min. Plasma was used to measure fasting blood glucose and fasting insulin levels. Measurement of fasting blood glucose was performed using a spectrophotometer with a commercially available kit (Dyasis). Fasting insulin levels were determined by ELISA (DRG Kit, cat #EIA2935). Buffy coat was used to extract DNA using a DNA isolation kit (GeneAid New Taipei City, Taiwan, cat #GEB100).

### 2.3. Genotyping

Genotyping was carried out using polymerase chain reaction–restriction fragment length polymorphism (PCR-RFLP) analysis. The PCR cycles were set as described in Table 1. For *ENPP1* rs1044498 SNP investigation, PCR products were obtained with the forward primer 5′-CTGTGTTCACTTTGGACATGTTG-3′ and reverse primer 5′-TCAAGGTCAGGTGCTCGTTGG-3′. The 207 bp amplified products were digested with AvaII restriction enzyme (Thermo Scientific, Waltham, MA, USA, cat #ER0381) for 16 h at 37 °C. The wild-type genotype (AA) was represented by a single band at 207 bp, heterozygous carriers (AC) were represented by three bands of 207, 148, and 59 bp, and homozygous individuals (CC) were represented by double bands of 148 and 59 bp.

To determine the rs1501299 SNP of *ADIPOQ*, we used the forward primer 5′- GGCCTCTTTCATCACAGACC-3′ and the reverse primer 5′-AGATGCAGCAAAGCCAAAGT -3′ to obtain amplified PCR products. The 196 bp amplified products were digested with the restriction enzyme MvaI (Thermo Scientific, cat #FD0554). The wild-type genotype (GG) was represented by double bands at 148 and 48 bp, heterozygous carriers (GT) were represented by three bands of 196, 148, and 48 bp, and non-risk alleles (TT) were represented by a single band at 196 bp. Genotype patterns of the *ENPP1* and *ADIPOQ* genes were analyzed by electrophoresis using 2% agarose gels.

### 2.4. Statistical Analysis

Normality and variance of data were tested by Kolmogorov–Smirnov and Levene tests, respectively. Differences between variables were compared using an unpaired *t*-test and one-way ANOVA. Log transformed data that did not follow normal distribution were computed using a Mann–Whitney test. Genotype and allele frequency were determined by a chi squared test. Fisher’s exact test was used if the expected value was less than 0.05. We used regression logistic analysis to determine the odds ratio (OR) and a *p* value was considered significant if *p* < 0.05.

## 3. Results

### 3.1. K121Q Polymorphism of ENPP1 Increased the Risk of Insulin Resistance

Clinical characteristics of the 100 subjects involved, including body weight and BMI, were significantly different between the case and control groups (Table 2). Fasting insulin and homeostatic model assessment of insulin resistance (HOMA-IR) were higher in the case group, as compared to the control group (Figure 1). With regard to the HOMA-IR value, 42 people with obesity, and 21 lean subjects, had insulin resistance.

In the next step, we performed PCR-RFLP to analyze the genotypes of the K121Q SNP of the *ENPP1* gene. All genotypes, including CC (risk alleles), AC, and AA (non-risk alleles), were detected in our study (Figure 2). Genotype AC/CC was found more frequently in the obesity group, and only one subject carrying the CC genotype was found, thus, we combined the data with genotype AC in the case group (Table 3). Allele distribution was not different between the lean and obese groups. We performed regression logistic analysis to calculate the OR of obesity and insulin resistance associated with the K121Q SNP of the *ENPP1* gene. Subjects carrying AC/CC genotypes possessed a 1.7-fold higher risk of obesity, but it did not reach the level of statistical significance (Table 3).

Obese individuals with the AA genotype were 5.5 (CI 2.0–14.3) times more likely to suffer insulin resistance (Table 4). Our study showed that subjects with obesity carrying the AC/CC genotypes possessed the highest risk of insulin resistance (OR = 6.6, 95% CI 1.3–34.5). These results were supported by higher values of fasting blood glucose in subjects carrying the AC/CC genotypes, regardless of their obesity status. Fasting insulin and HOMA-IR were significantly higher in obese but not in lean AC/CC genotype carriers. Meanwhile, BMI was not different between subjects carrying the AC/CC or AA genotypes (Figure 3). These data suggested that the K121Q SNP of the *ENPP1* gene increases the risk of insulin resistance in obese individuals.

### 3.2. Plasma Adiponectin Level Was Not Associated with the +276G > T SNP of ADIPOQ Gene

Next, we tested 110 individuals for the +276G > T SNP of the *ADIPOQ* gene (Figure 4). Clinical characteristics of 110 subjects are shown in Table 5. Subjects suffering from obesity had a slightly higher fasting blood glucose, and lower adiponectin levels, as compared to subjects with normal BMI (Figure 5). Distribution of allele and genotype frequency was not statistically different between the lean and obese groups (Table 6). There was no association between obesity and the +276G > T SNP of the *ADIPOQ* gene (OR = 1.08, 95% CI 0.5–2.3).

Finally, we investigated whether the +276G > T SNP of the *ADIPOQ* gene affected the plasma levels of adiponectin. There was no significant difference between the carriers of the three different genotypes at the aforementioned locus (Figure 6). These data suggest that there is no association between the +276G > T SNP of the *ADIPOQ* gene and plasma adiponectin levels, and confirm published data that obesity plays a major role in the reduction of adiponectin levels in Javanese people.

## 4. Discussion

Insulin resistance is the most common metabolic disturbance related to obesity. Our study confirmed that obese individuals possessed a higher risk of insulin resistance than lean individuals, in the Javanese population. Studies in the same ethnicity reported that obese individuals had higher levels of plasma insulin, leptin, and HOMA-IR as compared to lean individuals [22,23]. The protein level of ENPP1 was increased in the adipose tissue of individuals suffering from insulin resistance [24]. Adipose tissue ENPP1 protein expression positively correlated with plasma insulin levels when fasting, and after two hours following an oral glucose tolerance test (OGTT) [24]. Plasma glucose and insulin levels were also elevated when ENPP1 was overexpressed using an adenovirus vector, resulting in impaired insulin signaling [25]. Insulin-stimulated Akt phosphorylation in HuH7 human hepatoma cells was improved when *ENPP1* expression was silenced using siRNA [26]. Transgenic mice overexpressing the Q variant of ENPP1 had higher glucose and insulin levels, compared to wild-type mice [27]. The association of the K121Q polymorphism of the *ENPP1* gene with obesity and insulin resistance was investigated in various populations. Gonzáles-Sánches et al. reported that the presence of the Q variant of the *ENPP1* gene positively correlated with BMI and waist circumference, in Caucasians from Central Spain. They also reported that type 2 diabetic patients who carried the Q variant had higher BMI and leptin levels [28]. Another study in Denmark revealed that homozygous carriers of the QQ variant had a higher risk of being overweight, with OR 1.63 (95% CI, 1.09–2.46); however, there was no significant relationship between the presence of the Q variant and the development of type 2 diabetes or insulin resistance [29].

To our knowledge, our study reports for the first time that there is a positive association between the K121Q SNP of the *ENPP1* gene and increased risk of insulin resistance in the presence of obesity, in the Javanese population. Our finding is in agreement with a study which was carried out in the Moroccan population, reporting that the K121Q SNP was associated with type 2 diabetes in individuals suffering from obesity (OR = 1.55, 95% CI 1.16–2.07) [30]. This was also strengthened in the Italian white population; Baratta et al. reported that individuals carrying the homozygous QQ variant had a higher glucose profile during OGTT, and a significantly reduced insulinogenic index [31]. A meta-analysis in Asian and European populations revealed that the K121Q SNP of the *ENPP1* gene was associated with an increased susceptibility to diabetic kidney disease [32]. A previous study in a large French population reported that there was a positive association between the Q variant, and the risk of hyperglycemia (OR = 1.45) or type 2 diabetes (OR = 1.65), in subjects with a family history of type 2 diabetes [33].

The K121Q SNP results in a missense mutation that occurs in the somatomedin B-like domain of ENPP1. The somatomedin B-like domain enables ENPP1 to bind, and change the conformation of the insulin receptor [7]. Different charges of lysine (K) and glutamine (Q) variants contribute to the distinct affinity of ENPP1 to the insulin receptor in the cell membrane. Glutamine, which has a neutral charge, binds the insulin receptor with a higher affinity, compared to the positively charged lysine. Therefore, the Q variant increases the activity of the somatomedin B-like domain, resulting in impaired insulin signaling.

Our study did not significantly implicate a positive correlation between the K121Q SNP of the *ENPP1* gene and obesity. In accordance with our results, previous studies showed that there was no association between the K121Q polymorphism and obesity in the Chinese Han [13], French [33], and Taiwanese populations [34], the Caucasian population of the United Kingdom [35], and in Italian children [36]. On the other hand, other studies have reported that there was a positive correlation between the aforementioned SNP and obesity in the French population [33], and in European adults (meta-analysis) [37]. Conflicting results from various populations may have resulted from the diverse allele frequencies found in the assessed populations and ethnicities. The limitations of our study were the small sample size, wherein we could only find 20 individuals out of the total subjects who carried the Q variant (12 in the obese group and eight in the lean group), and the lack of assessment of lifestyle-related factors (e.g., smoking, physical activity). Future studies involving larger samples will be required to provide further insight into the effect of the K121Q SNP of the *ENPP1* gene, in the Javanese population.

Adiponectin is an adipokine that plays an important role in mediating the crosstalk between adipose tissue and other organs, including the liver, heart, pancreatic β-cells, kidney, and many other cell types in various tissues [38]. The release of adiponectin by adipocytes also mediates paracrine effects that allow the expansion of adipocyte number, up-regulation of genes involved in lipid metabolism, and reduction of pro-inflammatory regulators [39]. Two decades ago, mammalian adiponectin was purified, and the physiological role of adiponectin was investigated for the first time [40]. The blood glucose level of C57/BL6 mice was reduced after 4 h of adiponectin injection, and the decrease in blood glucose was not associated with any changes in the plasma insulin level. Furthermore, purified adiponectin reduced the blood glucose level in both type 1 and 2 diabetic mice [40]. Adiponectin is also essential for maintaining the expansion of healthy adipose tissue while rescuing ectopic lipid accumulation in animal models [41]. The overexpression of adiponectin in 3T3-L1 adipocytes led to an increase in adipogenesis and lipid storage [42].

In accordance with previous studies [14,15], we found that plasma adiponectin levels were lower in the obese group, rather than the lean group. A study on Indonesian men showed that male patients suffering from metabolic syndrome had lower adiponectin levels, compared to those without metabolic syndrome [43]. In adipocytes, several mechanisms, such as suppression of *ADIPOQ* transcription, increased oxidative stress, and subsequently decreased adiponectin translation, may contribute to the reduced levels of plasma adiponectin in obese individuals [17,44,45]. The *ADIPOQ* gene is a direct target of peroxisome proliferator-activated receptor (PPAR) γ, which also stimulates adiponectin multimerization and secretion, through enhancing the expression of several endoplasmic reticulum (ER) proteins [46,47,48]. Downregulation of PPARγ, in response to increased adiposity, subsequent inflammation, and ER stress, might contribute to the inhibition of adiponectin mRNA translation, and to the disruption of the biosynthesis of this multimeric adipokine [49,50]. The reduction of adiponectin mRNA expression in response to obesity might be regulated by the obesity regulatory element (ORE) sites; two of these were identified in the promoter region of the human *ADIPOQ* gene [51]. A similar mechanism was described in rodents with regard to the adipsin gene, which is expressed predominantly by adipocytes [52]. An ORE cis-element located upstream from the start of the adipsin gene was bound by proteins present less abundantly in nuclear extracts from obese, rather than lean, mice [53].

We found that the intronic +276G > T SNP of the *ADIPOQ* gene did not correlate with plasma adiponectin levels or obesity. Our finding is in accordance with previous studies in Jordanian [54], Chinese [55], Saudi [56], and young Taiwanese populations [57]. Of note, our study was carried out on a relatively low sample size that did not allow us to draw a more precise conclusion in this respect. Contrarily, other studies in various populations showed that the +276G > T SNP of the *ADIPOQ* gene was associated with susceptibility to type 2 diabetes, higher HOMA-IR, and higher adiponectin levels. A study in a Korean obese population revealed that GG homozygote carriers had lower HOMA-IR and elevated adiponectin levels, after a 12-week-long intervention of mild weight loss, but these changes were not found in subjects who carried the T allele [58]. It was also reported that TT homozygotes were associated with lower adiponectin levels, and a higher risk for obesity, insulin resistance, and parameters of metabolic syndrome in a Saudi Arabian population [59]. Another study that involved a Caucasian population in Romania reported that type 2 diabetic patients carrying TT homozygotes had higher plasma adiponectin levels than the GT or GG patients; however, the genotypes were not predictive for the development of type 2 diabetes [60]. A positive correlation between the presence of the T allele and a higher risk of obesity was found in the Indian Punjabi population [61]. Our study could not strengthen the previously found positive results in various populations with regard to the influence of the +276G > T SNP of the *ADIPOQ* gene on the plasma adiponectin levels, and on the development of obesity, however, we confirmed that obesity has a significant impact on reducing plasma adiponectin levels in the Javanese population.

## Figures and Tables

**Figure 1 life-11-00552-f001:**
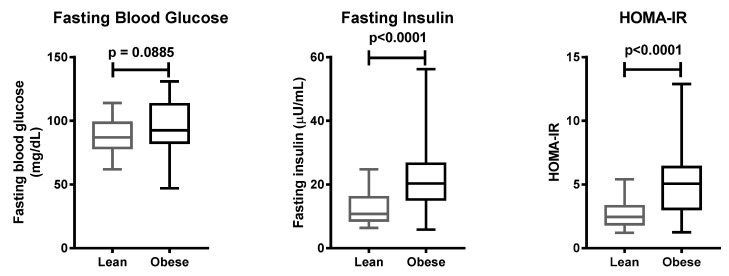
Clinical parameters of lean and obese subjects. Data are presented with boxplots, in which the boundary of the box closest to zero indicates the 25th percentile, the line within the box marks the mean, and the boundary of the box farthest from zero indicates the 75th percentile. Whiskers above and below the box indicate the minimum and maximum values. Statistical significance was analyzed by Mann–Whitney test.

**Figure 2 life-11-00552-f002:**
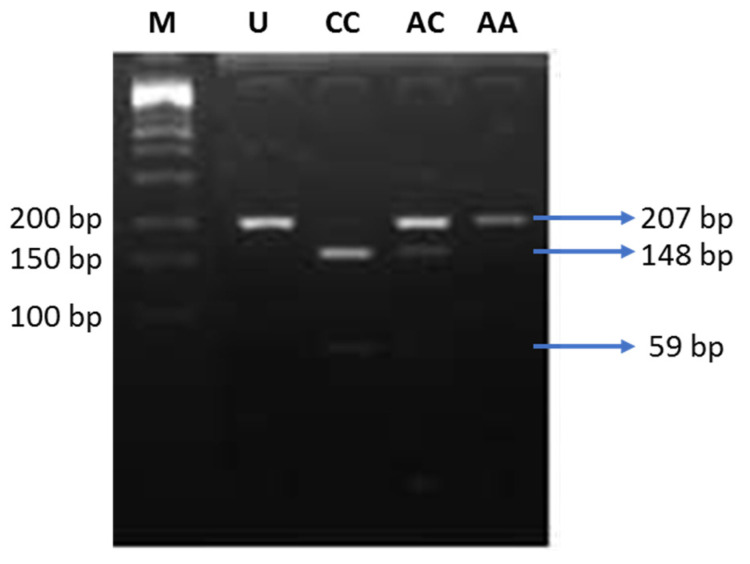
Electrophoresis of the digested products by AvaII. M: DNA marker 100 bp; U: uncut PCR product (207 bp); CC: risk alleles (148 and 59 bp); AC: heterozygous (207, 148, and 59 bp); AA: non-risk alleles (207 bp).

**Figure 3 life-11-00552-f003:**
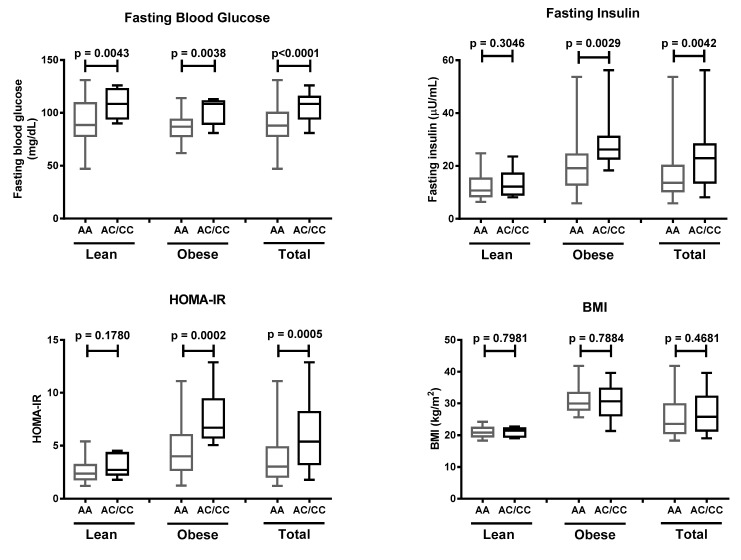
Fasting blood glucose, fasting insulin, homeostatic model assessment of insulin resistance (HOMA-IR) value, and body mass index (BMI) in lean, obese, and total subjects carrying AA and AC/CC genotypes at the K121Q SNP of *ENPP1* gene. Data are presented as in Figure 1. Statistical significance was analyzed by Mann–Whitney test.

**Figure 4 life-11-00552-f004:**
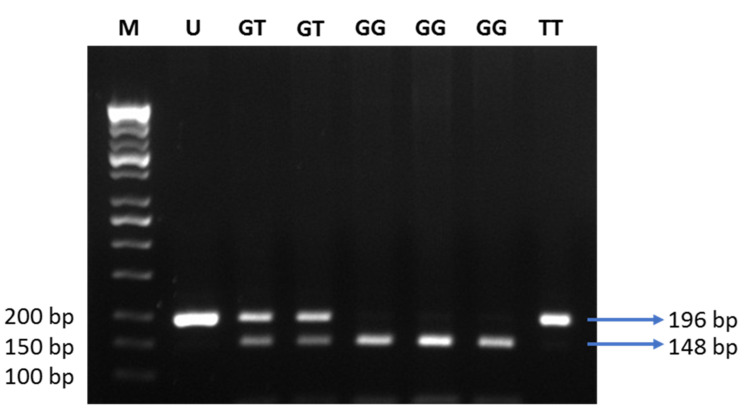
Electrophoresis of the digested products by MvaI. M: DNA marker 100 bp; U: uncut PCR product (196 bp); GG: risk alleles (148 and 48 bp); GT: heterozygous (196, 148, and 48 bp); TT: non-risk alleles (196 bp).

**Figure 5 life-11-00552-f005:**
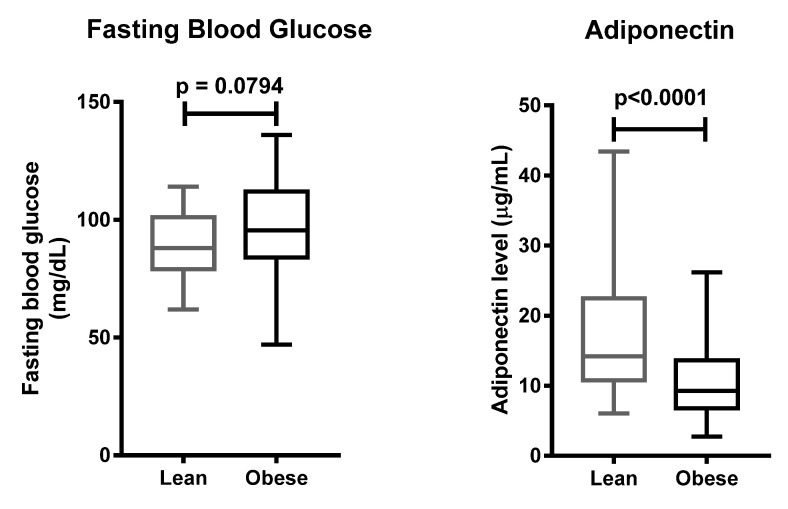
Fasting blood glucose and adiponectin levels of the control and case groups. Data are presented as in Figure 1 and Figure 3. Statistical significance was analyzed by Mann–Whitney test.

**Figure 6 life-11-00552-f006:**
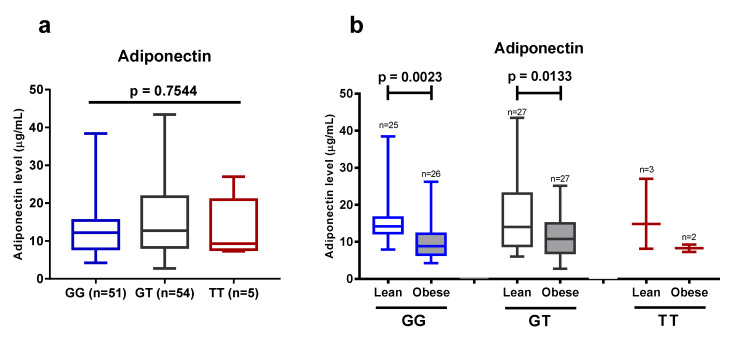
(**a**) Plasma adiponectin level in 110 subjects that carry different genotypes at the +276G > T SNP of *ADIPOQ* gene. (**b**) Adiponectin level in the lean and obese group with regard to the presence of different variants at the +276G > T SNP of *ADIPOQ* gene. Data are presented as in Figure 1, Figure 3, and Figure 5. Statistical significance was analyzed by one-way ANOVA (**a**) or unpaired Student’s *t*-test (**b**).

**Table 1 life-11-00552-t001:** The cycles of the polymerase chain reaction.

Step	Temperature	Time	Cycle
Initial denaturation	94 °C	5 min	1
Denaturation	94 °C	1 min	35
Annealing	55 °C	40 s	35
Extension	72 °C	40 s	35
Final extension	72 °C	10 min	1

**Table 2 life-11-00552-t002:** Characteristics of subjects involved in the study of K121Q SNP of *ENPP1* gene. Data are presented as mean ± SD. The normality of distribution of the data was analyzed by Kolmogorov–Smirnov test. Statistical significance was analyzed by Mann–Whitney test.

Variables	Lean	Obese	*p* Value
Females (*n*)	28	28	
Males (*n*)	22	22	
Age (years)	24.47 ± 5.00	24.66 ± 5.29	0.7331
Height (cm)	160.47 ± 8.79	162.27 ± 9.55	0.4165
Weight (kg)	52.68 ± 6.88	80.55 ± 14.17	**0.0042**
BMI (kg/m^2^)	21.0 ± 5.16	31.2 ± 4.5	**0.0027**

**Table 3 life-11-00552-t003:** Odds ratio (OR) of obesity in subjects carrying risk alleles at the K121Q SNP of *ENPP1* gene. OR was determined by using logistic regression analysis; CI: confidence interval.

		N	Obesity Status		OR (95% CI)	*p* Value
			Yes	No		
Genotype	AA	80	38 (76%)	42 (84%)	reference	0.320
	AC/CC	20	12 (24%)	8 (16%)	1.7 (0.6–4.5)	
Alleles	A	179	87 (87%)	92 (92%)	reference	0.888
	C	21	13 (13%)	8 (8%)	1.02 (0.6–2.0)	

**Table 4 life-11-00552-t004:** Odds ratio (OR) of insulin resistance risk in lean and obese groups carrying risk alleles at K121Q SNP of *ENPP1* gene. OR and *p* value were determined by logistic regression analysis and a control group carrying AA genotypes was considered as the reference; CI: confidence interval.

Group	Genotypes	Insulin Resistance Status		OR (95% CI)	*p* Value
		Yes	No		
Lean	AA	17 (40.5%)	25 (59.5%)	reference	
	AC	4 (50%)	4 (50%)	1.5 (0.3–6.7)	0.618
Obese	AA	30 (78.9%)	8 (21.1%)	5.5 (2.0–14.3)	**0.001**
	AC/CC	10 (83.3%)	2 (16.7%)	6.6 (1.3–34-5)	**0.025**

**Table 5 life-11-00552-t005:** Characteristics of subjects involved in the study of +276G > T SNP of *ADIPOQ* gene. Data are presented as mean ± SD. The normality of distribution of the data was analyzed by Kolmogorov–Smirnov test. Statistical significance was analyzed by Mann–Whitney test.

Variables	Lean	Obese	*p* Value
Females (*n*)	32	32	
Males (*n*)	23	23	
Age (Years)	22	22	0.5755
Height (cm)	160.3 ± 9.4	162 ± 9.92	0.3255
Weight (kg)	52.3 ± 7.80	80.8 ± 13.23	**<0.0001**
BMI (kg/m^2^)	21.04 ± 1.7	31.34 ± 4.4	**<0.0001**

**Table 6 life-11-00552-t006:** Odds ratio (OR) of obesity in subjects carrying risk alleles at the +276G > T SNP of *ADIPOQ* gene. OR and *p* value were determined by logistic regression analysis and a control group carrying GG genotypes was considered as the reference; CI: confidence interval.

		Obese (*n* = 55)	Lean (*n* = 55)	OR (CI 95%)	*p* Value
Genotypes	GG	26 (47.3%)	25 (45.5%)	reference	0.848
	GT/TT	29 (52.7%)	30 (54.5%)	1.08 (0.5–2.3)	
Alleles	G	79 (71.8%)	77 (70%)	reference	0.767
	T	31 (28.2%)	33 (30%)	1.09 (0.61–1.96)	

## Data Availability

The data presented in this study are available on request from the corresponding author. The data are not publicly available in accordance with consent provided by participants on the use of confidential data.

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
