# Peer review of "Influence of Single Nucleotide Polymorphism of ENPP1 and ADIPOQ on Insulin Resistance and Obesity: A Case—Control Study in a Javanese Population"

_life, 2021, doi:10.3390/life11060552_

Round 1
Reviewer 1 Report
Review comments:
Minor revision
The manuscript mainly descripted the impact of two SNPs, one is ENPP1-K121Q, one is ADIPOQ SNP +267G>T, on Insulin resistance and obesity-related metabolic traits, the major conclusion is ENPP1-K121Q variant increased the risk of insulin resistance in obese individuals in Javanese Population. The overall structure is clear, data is presented very well, the only concern is sample size is relatively quite small compared to many studies in the field with different populations.
Major:
- In figure3, the authors compared fasting blood glucose, fasting insulin and HOMA-IR level between genotypes, and the obese status was ignored. As shown in figure 1, there is significant difference between the lean and obese group, and merging individuals from different obese groups in the comparisons of figure3 may be problematic. It is better to firstly show a same pattern observed in different obese groups, and then merge the samples;
- In figure 6B, considering the large variance, I do not expect to see a significant difference between lean and obese individuals with one-way ANOVA. Please have a check, or provide more detailed explanation for the statistical test.
Minor:
- Reformat the Abstract as too many details numbers is not preferred, and make some conclusions at the end should be helpful for reader to catch the point;
- Add references for the statement in line 50-52;
- “codon 21” should be “codon 121” in line 61;
- Line 27 and line 134, “less than 5”, should be “less than .5”;
- Check the format in Table 3 for obese, like 24,66 should be 24.66;
- Add standard error for Weight in Table 5.
Author Response
The manuscript mainly descripted the impact of two SNPs, one is ENPP1-K121Q, one is ADIPOQ SNP +267G>T, on Insulin resistance and obesity-related metabolic traits, the major conclusion is ENPP1-K121Q variant increased the risk of insulin resistance in obese individuals in Javanese Population. The overall structure is clear, data is presented very well, the only concern is sample size is relatively quite small compared to many studies in the field with different populations.
We thank the Reviewer for the positive comments and useful suggestions which helped to improve the quality of the manuscript.
Major:
In figure3, the authors compared fasting blood glucose, fasting insulin and HOMA-IR level between genotypes, and the obese status was ignored. As shown in figure 1, there is significant difference between the lean and obese group, and merging individuals from different obese groups in the comparisons of figure3 may be problematic. It is better to firstly show a same pattern observed in different obese groups, and then merge the samples;
Answer: We agree that the previous presentation of the data was not accurate. Therefore, lean and obese groups are separately shown and the differences between the genotypes in each group are distinctly analyzed in the revised Figure 1.
In figure 6B, considering the large variance, I do not expect to see a significant difference between lean and obese individuals with one-way ANOVA. Please have a check, or provide more detailed explanation for the statistical test.
Answer: We agree that the previous description was not correct; one-way ANOVA was applied to analyze data displayed on Figure 6A, however unpaired t-test was used for 6B. This information is now added to the corresponding figure legend.
Minor:
Reformat the Abstract as too many details numbers is not preferred, and make some conclusions at the end should be helpful for reader to catch the point;
Answer: The Abstract was rephrased and the exact numerical values were removed.
Add references for the statement in line 50-52;
Answer: References 3 and 4 were included.
“codon 21” should be “codon 121” in line 61;
Line 27 and line 134, “less than 5”, should be “less than .5”;
Check the format in Table 3 for obese, like 24,66 should be 24.66;
Add standard error for Weight in Table 5.
Answer: The errors were corrected and the missing values were added.
Reviewer 2 Report
The Introduction is well-written and features an exact amount of information so as to familiarize the Readers with the problem, in context of genotypic variability of the studied polymorphisms and their clinical implication in different populations. The set of clinical markers discussed in this part is relevant to studies into obesity. No major grammar or spelling flaws were found in the text.
The information on the population sample is given in Section 2.1. However, the study did not take any lifestyle-related factors (such as exposition to cigarette smoke or active smoking etc.) into account. According to the current state of knowledge, obesity is a multifactorial disease. Therefore, this drawback of the study should be pointed out and discussed in the manuscript.
The genotyping section is thoroughly described, featuring all the vital information regarding PCR-RFLP and showing the bands associated with different genotypes (Figure 2 and Figure 4). However, the Statistical Analysis section lacks information regarding logistic regression and, in general, is insufficient for this manuscript to be of any use for planning future studies. Why is it assumed that the genotype variability is the only variable associated with the odds? This assumption increases the probability of false positive results. We could use many different variables to measure the OR of the incidence of obesity. However, they should be utilized in a multinomial model so as to account for other variables which might have had an influence on the OR and prediction of obesity. In the context of this study, at least two factors other than the SNPs could be utilized in the derivation of optimal logistic regression models; namely: age and sex; however, they could not be used in this study – due to insufficient sample size and balanced dataset (more-less equal count of males and females in the lean and obese groups, etc.) - this should be featured in the Discussion section. As the population sample is rather small, please try to analyse the OR in a multinomial logistic regression featuring both SNPs. Each factor should be analysed and described in a multinomial logistic regression, along with the interaction between these variables (or at least, please mention if it is significant). This multinomial model and two univariate models should be, then, described in terms of the goodness of fit (AIC, AICc, BIC etc.) and prediction power (learning AUC and testing AUC).
Please provide the exact p-values in the manuscript, as “n.s.” or “**” do not provide much information. A p-value of 0.058 could be “n.s.”, but by no means should it be neglected, as the sample size described in this study is rather low. Bar graphs should not be used, as they will not allow the Readers to analyse the value distribution. Please, use the boxplot (mean-median or median-mean type, depending on the performed analysis). It might be a good idea to feature a boxplot which would show the differences in standardized values of each variable, instead of many graphs showing the exact values of each variable (glucose, adiponectin, insulin, HOMA-IR, BMI), since the exact values of these parameters are already given in tables if the Readers would be interested to see them.
Overall, I highly recommend to publish this manuscript if these remarks would be addressed by the Authors in a major revision.
Author Response
The Introduction is well-written and features an exact amount of information so as to familiarize the Readers with the problem, in context of genotypic variability of the studied polymorphisms and their clinical implication in different populations. The set of clinical markers discussed in this part is relevant to studies into obesity. No major grammar or spelling flaws were found in the text.
We are grateful to the Reviewer for her/his insightful remarks which helped us to interpret our data more correctly.
The information on the population sample is given in Section 2.1. However, the study did not take any lifestyle-related factors (such as exposition to cigarette smoke or active smoking etc.) into account. According to the current state of knowledge, obesity is a multifactorial disease. Therefore, this drawback of the study should be pointed out and discussed in the manuscript.
Answer: This limitation of the study is now pointed out in the fourth paragraph of the Discussion.
The genotyping section is thoroughly described, featuring all the vital information regarding PCR-RFLP and showing the bands associated with different genotypes (Figure 2 and Figure 4). However, the Statistical Analysis section lacks information regarding logistic regression and, in general, is insufficient for this manuscript to be of any use for planning future studies. Why is it assumed that the genotype variability is the only variable associated with the odds? This assumption increases the probability of false positive results. We could use many different variables to measure the OR of the incidence of obesity. However, they should be utilized in a multinomial model so as to account for other variables which might have had an influence on the OR and prediction of obesity. In the context of this study, at least two factors other than the SNPs could be utilized in the derivation of optimal logistic regression models; namely: age and sex; however, they could not be used in this study – due to insufficient sample size and balanced dataset (more-less equal count of males and females in the lean and obese groups, etc.) - this should be featured in the Discussion section. As the population sample is rather small, please try to analyse the OR in a multinomial logistic regression featuring both SNPs. Each factor should be analysed and described in a multinomial logistic regression, along with the interaction between these variables (or at least, please mention if it is significant). This multinomial model and two univariate models should be, then, described in terms of the goodness of fit (AIC, AICc, BIC etc.) and prediction power (learning AUC and testing AUC).
Answer: In this study, we used chi square test to determine the odds ratio, simply because we aimed to observe the genotype and allele distribution adjusted to the obesity status. To our knowledge, we could not use multinomial logistic regression in our analysis because we only had two groups with regard to obesity status while the regression requires at least more than 2 groups. We also could not feature both SNPs in one regression model since the total number of subjects was different between these SNPs. We included 100 subjects in examining K121Q SNP of ENPP1 gene, while 110 subjects for +276G>T SNP of ADIPOQ gene.
Please provide the exact p-values in the manuscript, as “n.s.” or “**” do not provide much information. A p-value of 0.058 could be “n.s.”, but by no means should it be neglected, as the sample size described in this study is rather low. Bar graphs should not be used, as they will not allow the Readers to analyse the value distribution. Please, use the boxplot (mean-median or median-mean type, depending on the performed analysis). It might be a good idea to feature a boxplot which would show the differences in standardized values of each variable, instead of many graphs showing the exact values of each variable (glucose, adiponectin, insulin, HOMA-IR, BMI), since the exact values of these parameters are already given in tables if the Readers would be interested to see them.
Answer: We agree that the previous presentation of the data did not provide sufficient amount of information; therefore, we replaced the bar graphs on Figures 1, 3, 5, and 6 to boxplots. In addition, we indicated the exact p values in Tables 2-6 and in Figures 1, 3, 5, and 6.
Overall, I highly recommend to publish this manuscript if these remarks would be addressed by the Authors in a major revision.
We hope that after the revision the Reviewer will find the corrected manuscript acceptable for publication.
Round 2
Reviewer 2 Report
Thank You for the response. In the review I used the wrong term - I meant "multiple" logistic regression, not "multinomial". You have rightly pointed out that the count of patients was different in genotyping of the two SNPs. Trimming the data so as to equalize the counts would bias the results.
The manuscript in its current form is acceptable for publication.